# The Influence of Mistreatment by Patients on Job Satisfaction and Turnover Intention among Chinese Nurses: A Three-Wave Survey

**DOI:** 10.3390/ijerph17041256

**Published:** 2020-02-15

**Authors:** Lei Qi, Xin Wei, Yuhan Li, Bing Liu, Zikun Xu

**Affiliations:** 1School of Business Administration, Shandong University of Finance and Economics, Jinan 250014, China; leilasdu@163.com; 2School of Management, Shandong University, Jinan 250100, China; 17865196213@163.com (X.W.); hannlily@foxmail.com (Y.L.)

**Keywords:** mistreatment by patient, job satisfaction, turnover intention, work meaningfulness, emotional dissonance, hostile attribution bias

## Abstract

The affective event of mistreatment in the workplace has been recognized as an important factor influencing employee affect and behavior. However, few studies have logically explained and empirically clarified the link between mistreatment by patients and nurses’ job satisfaction and turnover intention. The current study aimed to explore the effects of mistreatment by patients on nurses’ job satisfaction and turnover intention through work meaningfulness and emotional dissonance, as well as the moderating role of hostile attribution bias. Using three-wave survey data collect from 657 nurses who worked in three hospitals in China, we found that mistreatment by patients had a negative effect on nurses’ job satisfaction through work meaningfulness, mistreatment by patients had a positive effect on nurses’ turnover intention through emotional dissonance. Furthermore, nurses’ hostile attribution bias acted as an effective moderator on the relationship. These findings help uncover the mechanisms and conditions in which mistreatment by patients influences nurses’ job satisfaction and turnover intention.

## 1. Introduction

According to the Chinese Medical Occupation Status White Paper issued by the Chinese Medical Doctor Association in 2018, 41.2% nurses have experienced mistreatment by a patient within the previous year [1]. This phenomenon (including verbal abuse, incivility, aggression) of patients mistreating nurses by making undeserved prejudicial statements or engaging in hurtful personal attacks can be described as mistreatment by patients [2,3]. As Grandey et al. found, mistreatment by service users is a common but problematic occurrence in the workplace [4]. Previous research has confirmed that an individual’s emotional wellbeing, attitudes, and performance can be compromised when they suffer mistreatment by service users [5,6,7]. Though many studies have explored mistreatment in the workplace, most concern customer mistreatment; few have paid attention to mistreatment by patients, which may reduce nurses’ job satisfaction and increase their turnover intention. 

Research has indicated that a healthy work environment is critical for nurses’ job satisfaction [8]. Based on affective events theory, mistreatment by patients will influence nurses’ job satisfaction through affecting their emotional state [9]. Work meaningfulness is an important factor in the subjective experience of how significant and intrinsically valuable people find their work to be [10,11]. Thus, we argue that mistreatment by patients is negatively related to employee job satisfaction through work meaningfulness.

Turnover intention has been investigated through the concept of emotional dissonance, which is considered a form of person-role conflict [12]. As Rafaeli et al. indicated, emotional dissonance occurs when an individual’s behavior conforms with organizational norms but is incongruent with their true feelings [13]. Several studies have suggested that individuals experiencing high emotional dissonance are more likely to quit [14,15]. However, when nurses experience mistreatment by patients, their professional behavior is incongruent with their true feelings, and thus emotional dissonance emerges. Despite a growing body of research focused on emotional dissonance, few researchers have explored the effect of mistreatment by patients on turnover intention through emotional dissonance. 

Wang et al. [7] suggested that employees’ attributions should be given attention in research on mistreatment by service users. It is plausible that nurses with high level of hostile attribution bias regarding patient mistreatment, they may not feel personally responsible for goal failure. Consequently, nurses may be less likely to gain meaningfulness from their work and more likely to feel emotional dissonance. Thus, examining this potential moderator provides an opportunity for us to understand the mechanism of mistreatment by patients. 

In sum, on the basis of affective events theory, we propose nurses’ work meaningfulness as an intermediate affective process that explains the relationship between mistreatment by patients and nurses’ job satisfaction. Meanwhile, we propose nurses’ emotional dissonance as an intermediate affective process that explains the relationship between mistreatment by patients and nurses’ turnover intention. Furthermore, we examine hostile attribution bias as a moderator of the mistreatment by patients-work meaningfulness/emotional dissonance association. The research model is presented in Figure 1.

## 2. Theoretical Review and Hypothesis Development

### 2.1. Mediating Role of Work Meaningfulness

Mistreatment by patients concerns rude, threatening, or aggressive behaviors directed at nurses [16], such as verbal insults or physical pushing. As a kind of job pressure, mistreatment influences employees’ recognition, emotional state, and behaviors [3]. Numerous studies have documented the negative effects of low-quality interpersonal treatment on service employees. For example, it can hinder an employee’s self-efficacy [17] and job performance [18], increase emotion exhaustion [2], and facilitate uncivilized behaviors [19]. 

Given that work has increasingly become a key domain from which people derive life meaningfulness [20], organizational researchers have increasingly turned their attention to work meaningfulness [21]. Martela et al. indicated that it has three dimensions: significance, broader purpose, and self-realization [21]. Significance is about how much intrinsic value people assign to or are able to find in their work. The broader purpose of work is to contribute to something greater than personal gain, such as society. Self-realization is about how much we are able to realize and express ourselves through our work; it can be gained by work autonomy [22]. Bailey et al. ´found that the interaction with other stakeholders can impact work meaningfulness [23]. Meanwhile, research has provided evidence that work meaningfulness is positively associated with work outcomes such as work engagement [24] and organization identification [25].

In this study, we anticipate that employees who have experienced more mistreatment by patients will perceive lower work meaningfulness. First, nurses engage in nursing because they believe the career have intrinsic value, that it is appreciated [26], but rough treatment by patients makes the job more stressful, which will reduce its perceived significance [27]. Second, one of the broader purposes of nursing is to ensure a better recovery for patients [28]. When confronted with mistreatment, nurses will feel they do not provide patients the best care [29] and doubt the added value of their work. They will perceive their work to be less meaningful. Third, nurses will feel less self-determination when they are mistreated [30]. If patients are depressed and act roughly with nurses, nurses will feel that their work is uncontrollable and difficult, and then doubt their own abilities [30]. Meanwhile, mistreatment by patients also indicates that nurses have established a bad connection with them and cannot meet the relationship needs of individual patients. With less autonomy and reduced social networks, nurses’ perceived work meaningfulness will decrease [21]. 

Affective events theory is an important theoretical framework that can be used to further explain the negative relationship between mistreatment by patients and nurses’ work meaningfulness. Affective events theory supposes that “work environment features will lead to the occurrence of positive or negative work events, and the experience of these work events will trigger individual emotional reactions, which will further affect individual attitudes and behaviors” [9]. We argue that mistreatment by patients represents a key affective event at work that elicits a strong emotional reaction [31,32]. More specifically, as a negative event, mistreatment by patients reduces nurses’ sense of social worth, belonging, and self-efficacy, which are important sources of work meaningfulness [22]. Therefore, mistreatment by patients is negatively related to work meaningfulness. 

We argue that as a result of reduced work meaningfulness, employees who are mistreated may experience a negative work state, that is, reduced job satisfaction. Job satisfaction can be defined as a pleasurable or positive emotional state resulting from the appraisal of one’s job or work experiences [33]. Research has shown that a healthy work environment is critical for nurses’ job satisfaction [8]. Drawing on affective events theory, we argue that mistreatment by patients is negatively related to employees’ job satisfaction through work meaningfulness.

Work meaningfulness answers the basic question, “Why do I work?” [34]. Employees with reduced work meaningfulness experience negative emotional states such as frustration, disappointment, and self-doubt [35]. When nurses’ needs (e.g., for self-actualization) cannot be met and their expectations of work have also been disenabled or hindered by patients, they will have less internal motivation to work [36]. Hence, nurses with low work meaningfulness are not able to maintain vitality and thrive at work, so they are less likely to have a satisfactory attitude toward work. 

Nurses who have experienced mistreatment by patients are more likely to doubt the value of nursing work for patients and society [29]. Meanwhile, their self-actualization expectations cannot be satisfied, and they cannot engage in personal growth and development [30]. Therefore, nurses will perceive low work meaningfulness. When nurses feel that their work is no longer meaningful enough, they are more likely to fall into depression and have lower job satisfaction. In other words, the more mistreatment by patients that nurses encounter, the less meaningful their work is, and the lower their job satisfaction. Furthermore, according to the meta-analysis of Humphrey et al., the experience of work meaningfulness might be the most critical factor of psychological state linking social and work characteristics and work outcomes [37]. Based on the above discussion, we propose the following hypothesis:
**Hypothesis** **1.**Work meaningfulness mediates the effect of mistreatment by patients on nurses’ job satisfaction.

### 2.2. Mediating Role of Emotional Dissonance

As a kind of emotional labor, emotional dissonance is defined as “the state that exists when there is a discrepancy between the emotional demeanor that an individual displays because it is considered appropriate, and the emotions that are genuinely felt but that would be inappropriate to display’’ [38], which can be induced by negative events. Grandey suggested that the source of these negative events might be clients, coworkers, supervisors, or personal situations [39]. Research has confirmed that unfair treatment by customers can lead to negative emotions in employees [2,40]. Hence, we predict that mistreatment by patients positively affects nurses’ emotional dissonance.

Affective events theory points out that emotional events in the work environment are the direct cause of emotional change [9]. Employees experience grievance, depression, and other negative emotions after experiencing mistreatment by patients [41]. Due to professional ethics, nurses tend to suppress negative emotions and show the positive attitude that the public desires. Meanwhile, nurses prefer to act kindly to have a good image in their organization and ensure future development opportunities, which produces more emotional dissonance. Stress is the result of an interaction between a person and their environment [42]. Dealing with mistreatment and dissatisfaction by patients increases the burden of nurses’ workload and makes them feel excessive work pressure [43,44]. Therefore, mistreatment by patients is positively related to emotional dissonance. 

Emotional dissonance, the incongruity between inner feelings and outer expressions, usually has a negative impact on employees. Mobley regarded turnover intention as an individual’s deliberate intention to leave an organization, resulting from dissatisfaction with their employment after working there for a certain period of time [45]. We propose that emotional dissonance prompts nurses’ turnover intention. 

Studies suggest that emotional dissonance is experienced as a person-role conflict [46]. Nurses try to suppress negative emotions and be positive employees, which consumes a lot of psychological resources and puts them in a state of tension. Long-term exposure to a high-stress environment makes employees more likely to leave because it is a quick way to escape from stressors [47]. In addition, based on the affective events theory, Grandey et al. found that negative emotional reactions can induce higher turnover intention [48]. Thus, emotional dissonance is correlated with turnover intention.

Studies have confirmed that the uncivilized behavior of customers increases the turnover intention of employees [49], but the internal mechanism of this process has not been thoroughly explored. Affective events theory proposes that work events first trigger individual emotional reactions, and further affect employees’ work attitudes and behaviors [9]. As an interpersonal pressure, mistreatment by patients causes negative emotions in employees [27]. Nurses are expected to hide their emotions when they are working [50], and thus they experience emotional dissonance. Individuals with emotional dissonance face greater pressure and have less of a sense of belonging in their organization [14]. Therefore, we propose the following hypothesis:
**Hypothesis** **2.**Emotional dissonance mediates the effect of mistreatment by patients on nurses’ turnover intention.

### 2.3. Moderating Role of Hostile Attribution Bias

Hostile attribution bias concerns individuals’ tendency to think that the behavior of others in uncertain situations is hostile [51]. Individuals with high hostile attribution bias will attribute the responsibility for mistakes to others or to circumstances, especially if the responsibility for the mistake is unclear [52]. Research has shown that trait hostility is implicated in negative outcomes [53], and that individuals with high hostile attribution bias are more likely to have negative psychological feelings after setbacks, such as anger and dissatisfaction [54]. Drawing on the appraisal theory of emotion, studies have shown that emotional reactions tend to emerge from the cognitive appraisal of a situation or causal attribution of events [55]. It can be seen that due to different attributional tendencies, different individuals may have different emotional responses to the same work event. Hence, we argue that hostile attribution bias will exacerbate the negative effects of mistreatment by patients on work meaningfulness and the positive effects of mistreatment by patients on emotional dissonance.

Nurses with high levels of hostile attribution bias are more likely to perceive less work meaningfulness when they suffer patients’ mistreatment. First, nurses with high hostile attribution bias tend to think that patients’ impolite behavior is revenge and that their dedicated service is not appreciated, which reduces their perceived work meaningfulness [52]. Second, after experiencing mistreatment by patients, nurses with high hostile attribution bias do not reflect on whether they have done something wrong, but regard patients’ aggression as gratuitous and incorrect [54]. Negative beliefs such as “no matter how hard I work, I cannot get recognition and fair treatment” and “this work is meaningless” will emerge. Furthermore, it has been found that individuals with high hostile attribution bias are more likely to take vindictive action after they suffer incivility [56]. 

By contrast, individuals with low levels of hostile attribution bias always blame themselves. When mistreatment by a patient occurs, they reflect on how their work is going wrong [53]. Or they think that the patient is in a bad mood because of their illness rather than specifically against them, when they should take responsibility for their own work and help the patient to recover. They think that work will bring benefits to others and society, and is meaningful [57]. In summary, nurses’ hostile attribution bias will influence the effect of mistreatment by patients on work meaningfulness. More formally, the following hypothesis is proposed:
**Hypothesis** **3.**Hostile attribution bias moderates the relationship between mistreatment by patients and work meaningfulness, such that the negative relationship is stronger when employees have high rather than low hostile attribution bias.

Furthermore, nurses with high hostile attribution bias tend to believe that mistreatment by patients is a kind of unwarranted retaliatory behavior, and such unfair interpersonal treatment reduces their perceived work meaningfulness, resulting in a low emotional work state, that is, a low degree of job satisfaction [32]. We thus hypothesize:
**Hypothesis** **4.**Hostile attribution bias moderates the relationship between mistreatment by patients and nurses’ job satisfaction through work meaningfulness, and the relationship is stronger when employees have high rather than low hostile attribution bias.

Similarly, we suggest that nurses with high hostile attribution bias are more likely to experience negative emotions, ultimately exacerbating the positive effect of mistreatment by patients on emotional dissonance. According to Hoobler et al., hostile attribution bias manifests more strongly in frustrating situations, so we speculate that it will also affect emotional reactions [58]. Nurses with high hostile attribution bias will classify patients’ aggression as an act of revenge when its intention is unclear [59]. As a result, they become angrier and even more vindictive [60]. However, these negative emotions can only be depressed because nurses avoid immoral retaliation against patients due to their morality or upward-mobility. As a result, they will experience more emotional dissonance. By contrast, nurses with low hostile attribution bias are genuinely sympathetic to their patients’ situation. When confronted with aggressive behavior from patients, such nurses are more understanding, calmer, more emotionally stable, and thus their emotional dissonance will drop. We hypothesize:
**Hypothesis** **5.**Hostile attribution bias moderates the relationship between mistreatment by patients and emotional dissonance, such that the positive relationship is stronger when employees have high rather than low hostile attribution bias.

When suffering mistreatment by patients, bias attribute their negative experiences to the patient [53], so they are more likely to experience emotional dissonance which will stay with them as long as they remain in the organization, thus enhancing these nurses’ intention to leave. In other words, mistreatment by patients will enhance the emotional disorder of nurses with high hostile attribution, and hostile attribution bias will enhance this positive relationship, thus further enhancing the effect of mistreatment by patients on turnover intention through emotional dissonance. Following the above discussion, we believe that nurses’ hostile attribution bias plays a moderating role in the path of “mistreatment by patient–emotional dissonance–turnover intention.” We thus hypothesize:
**Hypothesis** **6.**Hostile attribution bias moderates the relationship between mistreatment by patients and nurses’ turnover intention through emotional dissonance, and the relationship is stronger when employees have high rather than low hostile attribution bias.

## 3. Methods

### 3.1. Study Design and Participants

To alleviate the potential common method bias concern [61], the study adopted a time-lagged three-wave survey design (data were collected from November 2016 to January 2017) to examine the mediating effect of work meaningfulness in the relationship between mistreatment by patients and nurses’ job satisfaction, the mediating effect of emotional dissonance in the relationship between mistreatment by patients and nurses’ turnover intention, as well as the moderating effect of hostile attribution bias.

Nurses from three hospitals, two in Jinan and one Taiyuan, China participated in the survey. Hospital managers and human resource supervisor authorized this survey. We confirmed that each respondent is willing to accept the questionnaire survey before we delivering the questionnaires. All the participants were completely free to join or drop out of the survey. A total of 1200 nurses were randomly selected to participate in the survey. They received 5–20 RMB (about 1–3 US dollars) after completing each questionnaire. The researchers visited these hospitals and managed the survey processes following the same protocol: we explained the purpose of the study, assured confidentiality, emphasized the importance of honest answers for scientific research before distributing the survey questionnaires, and took back the completed questionnaire directly on the spot. 

The survey questionnaires were coded before being distributed. At time one (T1), we administered questionnaires to 1200 nurses, who were asked to provide their demographic information (e.g., age, gender, and tenure), and information relating to mistreatment by patients and hostile attribution bias. We received usable responses from 1067 employees, giving a response rate of 88.9%. At time two (T2, one month after T1), the same 1067 employees were targeted to report on their work meaningfulness and emotional dissonance. We received 921 usable responses, giving a response rate of 77%. At time three (T3, one month after T2), the 921 employees who completed the T2 survey were asked to report their job satisfaction and turnover intention. Since six of these people had left during this period, 915 employees received the questionnaires. We obtained 739 responses (a response rate of 61.58%) and a final sample of 657 observations after excluding missing data (valid response rate was 54.75%). Among these 657 nurses, 74.58% were female with an average age of 30.29 years (SD = 7.13 years) and an average organizational tenure of 8.01 years (SD = 7.36 years), while 25.42% were male with an average age of 32.63 years (SD = 7.59 years) and an average organizational tenure of nine years (SD = 7.82 years).

### 3.2. Measurement Scales

We followed the back-translation procedure recommended by Brislin to translate the measures [62]. A management professor fluent in both English and Chinese translated the items from English to Chinese, and then another bilingual management professor translated them from Chinese back into English. Discrepancies in the translation were resolved through discussion. A few participants from the participating hospitals were consulted to guarantee that the items could be generalized to the study context [63]. We used a seven-point Likert scale to assess each measure (1 = “totally disagree” to 7 = “totally agree”).

Mistreatment by patients (T1). We adopted the seven-item mistreatment by patients scale from Cortina et al. [64]. Items include: “Patients doubted my judgment on a matter over which I have responsibility”, “Patients made unwanted attempts to draw me into a discussion of personal matters”, “Patients addressed me in an unprofessional term, either publicly or privately”, “Patients put me down or was condescending to me”, “Patients paid little attention to my statement or showed little interest in my opinion”, “Patients made demeaning or derogatory remarks about me”, and “Patients yelled at me”. Cronbach’s alpha for this measure was 0.982. 

Hostile attribution bias (T1). We adopted six items from Adams et al. to assess hostile attribution bias [53]. Items include: “Most people are honest chiefly through fear of being caught” and “I commonly wonder what hidden reason another person may have for doing nice to me”. Cronbach’s alpha for this measure was 0.957.

Work meaningfulness (T2). We used Duchon et al.’s seven-item scale to measure work meaningfulness [65]. Items include: “I see a connection between my work and the larger social good of my community” and “My spirit is energized by my work”. Cronbach’s alpha for this measure was 0.958.

Emotional dissonance (T2). We measured emotional dissonance using a five-item scale developed by Kumar et al. [66]. Items include: “I have to cover up my true feelings when dealing with patients”, “I spend most of my workday faking positive emotions”, and “In my work I have to be friendly with patients, even when I do not want to”. Cronbach’s alpha for this measure was 0.900.

Job satisfaction (T3). We adapted the five-item scale developed by Wright et al. and Braun et al. to measure job satisfaction [67,68]. Items include: “How satisfied are you with the working conditions?” and “How satisfied are you with support of your professional career?” The Cronbach’s alpha for this scale was 0.952.

Turnover intention (T3). A three-item scale developed by Knudsen et al. (2006) was adapted to measure employee’s turnover intention [69]. Items include: “As soon as I can find a better job, I will leave this hospital”, “I am actively looking for a job at another hospital”, and “I am seriously thinking of quitting my job”. The Cronbach’s alpha for this scale was 0.892.

Control variables. Research has indicated that employee demographics are likely correlated with job satisfaction [70] and turnover intention [71]. We thus followed previous research [72] to control gender, age, and organizational tenure for their potential effects.

### 3.3. Data Analysis

We used SPSS 23.0 to implement reliability and descriptive statistics and employed Mplus 7.4 to implement CFA before testing our hypotheses. We used a bootstrapping approach, including 95% confidence intervals (CI) using 2000 bootstrap samples in Mplus 7.4 to test our hypotheses. 

## 4. Results

### 4.1. Descriptive Statistics Correlations

In this study, we used SPSS20.0 and Mplus7.4 to analyze the data. Table 1 presents the means, standard deviations, and correlations of the key variables in our study. The results indicate that employee age (r = 0.094, *p* < 0.05) and tenure (r = 0.110, *p* < 0.01) both have positive correlations with emotional dissonance. Mistreatment by patients was negatively correlated with work meaningfulness (r = –0.172, *p* < 0.01) and positively correlated with emotional dissonance (r = 0.430, *p* < 0.01). Work meaningfulness was positively correlated with job satisfaction (r = 0.642, *p* < 0.01). Emotional dissonance was positively correlated with turnover intention (r = 0.392, *p* < 0.01). These results provide initial support for our hypotheses.

### 4.2. Confirmatory Factor Analyses

Since the data were collected from one source, we conducted confirmatory factor analyses to evaluate the possibility of same-source bias by testing whether these variables were distinct from each other [73]. We examined our hypothesized six-factor model and compared it with other alternative models. The result, as shown in Table 2, suggested that the proposed six-factor model fit the data best: χ^2^ = 279.726, df = 89, *p* < 0.01, Confirmatory Fit Index (CFI) = 0.985, Tucker-Lewis Index (TLI) = 0.979, root mean square error of approximation (RMSEA) = 0.057, and standardized root-mean-square residual (SRMR) = 0.027. These findings suggest empirical distinctions among the variables of this research.

### 4.3. Hypothesis Testing

We conducted a multilevel structural equation modeling framework with Mplus 7.4 to test our indirect effects, moderated effects, and moderated mediation effects [74]. Table 3 provides the unstandardized coefficient estimates, standard errors, and 95% confidence intervals (CIs) for the research model. As shown in Table 3, the indirect effect of mistreatment by patients on job satisfaction through work meaningfulness was significant (β = −0.087**, S.E. = 0.023, 95% [CI] = [−0.135, −0.043]), and so Hypothesis 1 was supported. The indirect effect of mistreatment by patients on turnover intention through emotional dissonance was significant (β = −0.067**, S.E. = 0.020, 95% [CI] = [0.032, 0.123]), so Hypothesis 2 was supported.

Table 4 and Table 5 present the results of the moderated and moderated mediation models. Table 4 shows that the interaction of mistreatment by patients with hostile attribution was negatively significant in predicting work meaningfulness (*p* < 0.01). Table 5 shows the interaction of mistreatment by patients with hostile attribution was positively significant in predicting emotional dissonance (*p* < 0.05). To provide additional evidence for the hypothesis, based on Aiken and West’s (1990) study, we used simple slope analyses at high and low levels of hostile attribution bias (1 SD above and below the mean). As shown in Figure 2, the negative relationship between mistreatment by patients and work meaningfulness was weaker among nurses with a higher level of hostile attribution bias. Hypothesis 3 was not supported. As shown in Figure 3, the relationship between mistreatment by patients and emotional dissonance was stronger among nurses with a higher level of hostile attribution bias. Hypothesis 5 was supported.

Next, we examined the conditional indirect effects of mistreatment by patients on job satisfaction through work meaningfulness, as well as the conditional indirect effects of mistreatment by patients on turnover intention through emotional dissonance at two values of hostile attribution bias (one SD below the mean and one SD above the mean). As shown in Table 4, the conditional indirect effect for mistreatment by patients was significant for low hostile attribution bias (*p* < 0.01), but not for high hostile attribution bias (*p* > 0.05). Taken together, the results indicate that hostile attribution bias negatively moderated the indirect relationship between mistreatment by patients and job satisfaction through work meaningfulness. Hypothesis 4 was not supported. As shown in Table 5, the conditional indirect effect for mistreatment by patients was significant for high hostile attribution bias (*p* < 0.05), but not for low hostile attribution bias (*p* > 0.05). Taken together, the results indicate that hostile attribution bias moderated the indirect relationship between mistreatment by patients and turnover intention through emotional dissonance. Hypothesis 6 was supported.

## 5. Discussion

The current study represents one of the first attempts to use affective events theory to explain nurses’ job satisfaction and turnover intention triggered by mistreatment by patients. We found that when a nurse experienced more mistreatment by patients, they perceived their work to be less meaningful and experienced more emotional dissonance, which in turn led to lower levels of job satisfaction and higher levels of turnover intention. In addition, we found that nurses’ hostile attribution bias moderated the effects of mistreatment by patients on work meaningfulness and emotional dissonance, such that the positive effect of mistreatment by patients on emotional dissonance was stronger among those who had higher levels of hostile attribution bias. What surprised us was that the negative effect of mistreatment by patients on work meaningfulness was stronger among those who had lower levels of hostile attribution bias, while the effect of mistreatment by patients on work meaningfulness was positive among those who had higher levels of hostile attribution bias. Because when nurses with lower levels of hostile attribution bias suffered mistreatment, they think that patients do this because their work is not recognized by their patients, which further decreases their work meaningfulness. On the contrary, when nurses with higher levels of hostile attribution bias suffered mistreatment, nurse believe they just did what they have to do, they may treat this mistreatment as malice on the part of the patients [57], instead, which inspires their inner stubbornness, which may further increase their work meaningfulness.

### 5.1. Theoretical Contributions and Practical Implications

Researchers broadly agree that mistreatment by service users influences service providers’ recognition, emotional state, and behaviors [75]. Existing studies, however, do not provide any reliable theoretical framework for conducting research into mistreatment in the nursing context. The current study has focused on an important, yet mostly unexplored issue relating to mistreatment by patients in the nursing context, a context in which research on mistreatment has not been carried out in the past. 

This study extends research on mistreatment by patients to the domain of nurses’ job satisfaction and turnover intention, which are increasingly recognized as critical factors for employee retention [76]. The mistreatment literature predominantly focuses on employees’ emotional and retaliatory variables as the outcomes of customer mistreatment [7,77]. A key contention in mistreatment studies is that customer mistreatment enhances employees’ negative emotions [78]. We propose that, in the nursing context, this contention hinges on the critical role played by employee experiences of mistreatment by patients in reducing employees’ perceived work meaningfulness, the source of job satisfaction, as well as increasing employees’ emotional dissonance, the source of turnover intention, which was consistent with previous research [79], mistreatment by service receiver has negative influence on service provider’s emotions and actions. 

Our study answers the calls of previous research to study and attest to the importance of hostile attribution in the relationship between mistreatment by patients and work meaningfulness/emotional dissonance. Recent mistreatment research shows that the moderating effect of hostile attribution bias on the associations between mistreatment and emotional reactions is not yet documented [7]. In line with attribution theory [80], this study advances an attribution model on mistreatment by patients and demonstrates that it will have a weaker negative effect on work meaningfulness (a stronger positive effect on emotional dissonance) when hostile attribution bias is high, rather than low.

The current findings have valuable implications for nurse managers. Our study shows that mistreatment by patients is significant in lowering nurses’ job satisfaction and increasing their turnover intention while previous study showed that the safe work environments which are devoid of violence may enhance nurses’ job satisfaction [81]. One strategy would be to take steps to prevent mistreatment by patients from occurring in the first place. Hospitals might use signs to remind patients to treat nurses with dignity and respect and to institute a zero-tolerance policy toward mistreatment by patients [82]. This strategy might not only help to reduce mistreatment by patients but also show that hospitals care about treating nurses with dignity and respect. 

Our findings indicate that hostile attribution bias moderates the relationship between mistreatment by patients and work meaningfulness/emotional dissonance, which also provides a new insight into how managers can approach this problem. Previous research has suggested that hostile attribution bias is changeable [51]. Managers should discourage nurses’ hostile attribution bias; it is plausible that when nurses attribute mistreatment to reasons other than hostility, there is no need to respond negatively. Moreover, Yagil and Dayan’s [83] findings extended extant research on the causes of mistreatment against nurses by highlighting a tendency to view certain circumstances as justifying such behavior; that if the nurse was not caring enough, or did not communicate clearly enough, the patient had the right to get aggressive. Or that the patient is just in great distress, enough to justify mistreatment, which one needs to be trained to tolerate. There has to be a shift in how we perceive or tolerate or justify mistreatment, so that organizations and the public can be educated to embrace fundamental changes on what is acceptable behavior towards healthcare providers. 

In addition, organizations can help nurses to deal with mistreatment by patients and associated resource loss. Previous research has suggested that an individual’s perceived social support at work can buffer the negative effects of mistreatment [84]. Further support, such as supervisor or organizational support, should be provided to nurses to help them gain and maintain resources to avoid emotional dissonance and the loss of work meaningfulness [85].

### 5.2. Limitations and Future Research Directions

Although this study makes numerous theoretical and practical contributions, it also has several limitations that are worthy of exploration in future research. First, although we tested our hypotheses using a relatively large and appropriate sample, our sample data were still limited by the fact that they were drawn from only two regions. This might raise concerns regarding the generalizability of our results. Future research conduct in different regions with more diverse employees would provide greater insight into the relationships we have proposed and also improve external validity.

Second, although we took special precautions to collect data in three waves, we still tested our research model based on cross-sectional data, which cannot rule out the issue of reverse causality. Employees with lower perceived work meaningfulness and more emotional dissonance might cause greater levels of mistreatment by patients. Thus, it is necessary to use longitudinal data in future which not only rule out common method variance, but also uncover the dynamic influence of mistreatment by patients. Although we took nurses’ gender, age, organizational tenure as controls, there are many other factors like education, type of work, and additional qualifications may influence nurses’ job satisfaction and turnover intention. We call for future research control more factors for their potential effects. 

Third, we focused on hostile attribution bias as a moderator to clarify the boundary effect of mistreatment by patients on work meaningfulness and emotional dissonance. Given that mistreatment by patients has an irreversible negative influence on employees and causes them physical and mental harm [86], we call for future research to investigate positive individual characteristics (i.e., core self-evaluation) and positive organizational level conditions (i.e., caring climate, leader support) that may buffer service employees’ emotional reactions to mistreatment by patients. 

## 6. Conclusions

Mistreatment by service users is a common but problematic occurrence in the workplace. The affective event of mistreatment by patients in the workplace has been recognized as an important factor influencing nurses affect and behavior. The current study revealed the mechanisms by which work meaningfulness and emotional dissonance that are evoked from mistreatment by patients may be related to nurses’ job satisfaction and turnover intention, as well as the moderated role of hostile attribution bias. Thus, a zero-tolerance policy toward mistreatment by patients and a mentoring programme toward hostile attribution bias are important to improve nurses’ job satisfaction and reduce nurses’ turnover intention. 

## Figures and Tables

**Figure 1 ijerph-17-01256-f001:**
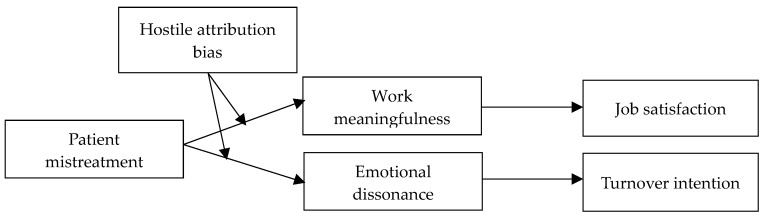
Research model.

**Figure 2 ijerph-17-01256-f002:**
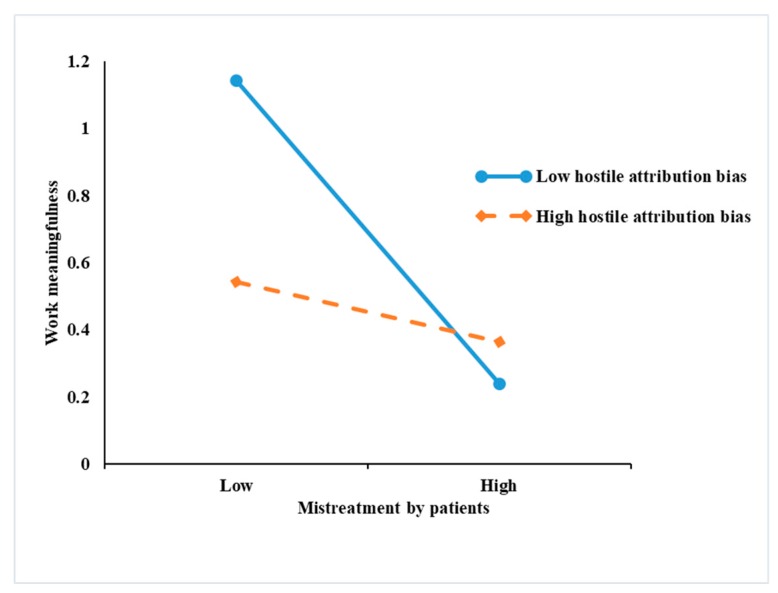
Interactive effects of mistreatment by patients and hostile attribution bias on work meaningfulness.

**Figure 3 ijerph-17-01256-f003:**
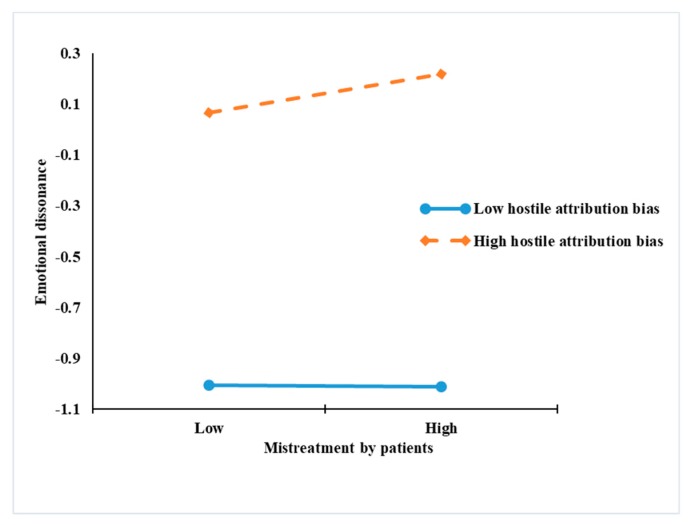
Interactive effects of mistreatment by patients and hostile attribution bias on emotional dissonance.

**Table 1 ijerph-17-01256-t001:** Descriptive statistics and correlations.

	M	S.D.	1	2	3	4	5	6	7	8
1.MP	2.170	1.429								
2.WM	5.114	1.139	−0.172 **							
3.JS	5.042	1.154	−0.118 **	0.642 **						
4.ED	3.455	1.348	0.430 **	−0.135 **	−0.110 **					
5.TI	2.908	1.434	0.610 **	−0.260 **	−0.301 **	0.392 **				
6.HAB	2.930	1.358	0.593 **	−0.185 **	−0.202 **	0.632 **	0.596 **			
7.Gender	1.746	0.436	−0.129 **	0.046	0.050	−0.039	−0.146 **	−0.073		
8.Age	30.887	7.311	0.070	−0.072	−0.056	0.094 *	0.004	0.090 *	−0.139 **	
9.Tenure	8.258	7.485	0.066	−0.042	−0.032	0.110 **	−0.008	0.121 **	−0.058	0.927 **

Note: N = 657; ** *p* < 0.01; * *p* < 0.05. MP = Mistreatment by patient, WM = Work meaningfulness, JS = Job satisfaction, ED = Emotional dissonance, TI = Turnover intention, HAB = Hostile bias attribution.

**Table 2 ijerph-17-01256-t002:** Results of the confirmatory factor analysis.

Model	$x^{2}$	df	CFI	TLI	RMSEA	SRMR
Six-factor model: MP, HAB, WM, JS, ED, TI	279.726	89	0.985	0.979	0.057	0.027
Five-factor model: MP + HAB, WM, JS, ED, TI	2246.489	94	0.827	0.779	0.187	0.116
Five-factor model: MP, HAB, WM + JS, ED, TI	1017.690	94	0.926	0.905	0.122	0.052
Five-factor model: MP, HAB, WM, JS, ED + TI	1200.254	94	0.911	0.886	0.134	0.081
One-factor model: MP + HAB + WM + JS + ED + TI	7765.559	104	0.383	0.289	0.335	0.199

Note: N = 657.

**Table 3 ijerph-17-01256-t003:** Results of hypotheses testing.

	Path	Estimates	S.E.	95% CI
Mediating effects	MP→WM→JS	−0.087 **	0.023	[−0.135, −0.043]
	MP→ED→TI	0.067 **	0.020	[0.032, 0.123]

Note: N = 657; ** *p* < 0.01.

**Table 4 ijerph-17-01256-t004:** Direct, indirect, and total effects at different levels of hostile attribution bias.

Model	Mistreatment by Patients → Work Meaningfulness → Job Satisfaction			
	PMX	Direct Effect (PYX)	Indirect Effect (PYMPMX)	Total Effects (PYX + PYMPMX)
High hostile attribution bias	−0.024	0.063 *	−0.013	0.050
Low hostile attribution bias	−0.516 **	−0.140 **	−0.283 **	−0.423 **
The difference	0.492 **	0.203 **	0.269 **	0.473 **

Note: N = 657; ** *p* < 0.01; * *p* < 0.05.

**Table 5 ijerph-17-01256-t005:** Direct, indirect, and total effects at different levels of hostile attribution bias.

Model	Mistreatment by Patients → Emotional Dissonance → Turnover Intention			
	PMX	Direct Effect (PYX)	Indirect Effect (PYMPMX)	Total Effects (PYX + PYMPMX)
High hostile attribution bias	0.091 *	0.406 **	0.05 *	0.455 **
Low hostile attribution bias	−0.018	0.332 **	−0.010	0.323 **
The difference	0.108 *	0.073	0.059 *	0.133 *

Note: N = 657; ** *p* < 0.01; * *p* < 0.05.

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
