# Peer review of "The Influence of Mistreatment by Patients on Job Satisfaction and Turnover Intention among Chinese Nurses: A Three-Wave Survey"

_ijerph, 2020, doi:10.3390/ijerph17041256_

Round 1

Reviewer 1 Report

INTRODUCTION

Paragraph 2 is part of the introduction and is very long. The authors should unify and simplify this section a little.

METHODS

The authors have not described the design in this section.

The statitic analysis performed should be described in this section.

DISCUSSION

The first paragraph repeats information from the results and does not justify the findings by comparing it with other research.

The authors have used few references in discussion and many are obsolete.

REFERENCES

Many bibliographies are obsolete. The bibliographic citations used are more than 5 years old (77,9%). The authors must update and arrange the bibliography.

Author Response

Response Letter 

The influence of mistreatment by patients on job satisfaction and turnover intention among Chinese nurses: A three-wave survey

Dear Editor and Reviewers,

Thanks very much for providing us with an opportunity to revise and resubmit our manuscript. We sincerely thank you for your kind and helpful comments. We have deeply considered each of the comments and carefully revised the manuscript. In this revised version, we hope our major changes can significantly improve our manuscript and thus meet your expectations. For your convenience, we provide specific response to each of the comments by the reviewers below.

Reviewer #1:

Comments and Suggestions for Authors

Response:

Thank you for your suggestion. We have carefully thought every piece of your advice and try our best to solve these issues, hoping to improve our manuscript and meet your requirements. Thanks for your suggestions again.

INTRODUCTION

Paragraph 2 is part of the introduction and is very long. The authors should unify and simplify this section a little.

Response:

Thank you for your suggestion. In the revised manuscript, we unified and simplified this section. Specific as follow:

“According to the Chinese Medical Occupation Status White Paper issued by the Chinese Medical Doctor Association in 2018, 41.2% nurses have experienced mistreatment by a patient within one year . This phenomenon (including verbal abuse, incivility, aggression) of patients mistreating nurses by making undeserved prejudicial statements or engaging in hurtful personal attacks can be described as mistreatment by patients [1,2]. As Grandey et al. found, mistreatment by service users is a common but problematic occurrence in the workplace [3]. Previous research has confirmed that an individual’s emotional well-being, attitudes, and performance can be compromised when they suffer mistreatment by service users [4–6]. Though many studies have explored mistreatment in the workplace, most concern customer mistreatment; few have paid attention to mistreatment by patients, which may reduce nurses’ job satisfaction and increase their turnover intention.

Research has indicated that a healthy work environment is critical for nurses’ job satisfaction [7]. Based on affective events theory, mistreatment by patients will influence nurses’ job satisfaction through affecting their emotional state [8]. Work meaningfulness is an important factor in the subjective experience of how significant and intrinsically valuable people find their work to be [9,10]. Thus, we argue that mistreatment by patients is negatively related to employee job satisfaction through work meaningfulness.

Turnover intention has been investigated through the concept of emotional dissonance, which is considered a form of person-role conflict [11]. As Rafaeli et al. indicated, emotional dissonance occurs when an individual’s behavior conforms with organizational norms but is incongruent with their true feelings [12]. Several studies have suggested that individuals experiencing high emotional dissonance are more likely to quit [13,14]. However, when nurses experience mistreatment by patients, their professional behavior is incongruent with their true feelings, and thus emotional dissonance emerges. Despite a growing body of research focused on emotional dissonance, few researchers have explored the effect of mistreatment by patients on turnover intention through emotional dissonance.

Wang et al. [6] suggested that employees’ attributions should be given attention in research on mistreatment by service users. It is plausible that nurses with high level of hostile attribution bias regarding patient mistreatment, they may not feel personally responsible for goal failure. Consequently, nurses may be less likely to gain meaningfulness from their work and more likely to feel emotional dissonance. Thus, examining this potential moderator provides an opportunity for us to understand the mechanism of mistreatment by patients.

In sum, on the basis of affective events theory, we propose nurses’ work meaningfulness as an intermediate affective process that explains the relationship between mistreatment by patients and nurses’ job satisfaction. Meanwhile, we propose nurses’ emotional dissonance as an intermediate affective process that explains the relationship between mistreatment by patients and nurses’ turnover intention. Furthermore, we examine hostile attribution bias as a moderator of the mistreatment by patients-work meaningfulness/emotional dissonance association. The research model is presented in Figure 1.”

METHODS

The authors have not described the design in this section.

Response:

Thank you for your comment. In the revised manuscript, we added the design in this section. Specific as follow:

“3.1. Study Design and Participants and Procedures

To alleviate the potential common method bias concern[60], the study adopted a time-lagged three-wave survey design (data were collected from November 2016 to January 2017) to examine the mediating effect of work meaningfulness in the relationship between mistreatment by patients and nurses’ job satisfaction, the mediating effect of emotional dissonance in the relationship between mistreatment by patients and nurses’ turnover intention, as well as the moderating effect of hostile attribution bias.”

The statitic analysis performed should be described in this section.

Response:

Thank you for your comment. In the revised manuscript, we added the design in this section. Specific as follow:

“3.3. Data Analysis

We used SPSS 23.0 to implement reliability and descriptive statistics and employed Mplus 7.4 to implement CFA before testing our hypotheses. We used a bootstrapping approach, including 95% confidence intervals (CI) using 2000 bootstrap samples in Mplus 7.4 to test our hypotheses.”

DISCUSSION

The first paragraph repeats information from the results and does not justify the findings by comparing it with other research.

The authors have used few references in discussion and many are obsolete.

Response:

Thank you for your comment. The first paragraph is a conclusion of the study, more discussions are in the section of 5.1. Theoretical Contributions and Practical Implications.

In the revised manuscript, we added more justification to compare it with other research, some new references were included. Specific as follow:

“Researchers broadly agree that mistreatment by service users influences service providers’ recognition, emotional state, and behaviors[74]. Existing studies, however, do not provide any reliable theoretical framework for conducting research into mistreatment in the nursing context. The current study has focused on an important, yet mostly unexplored issue relating to mistreatment by patients in the nursing context, a context in which research on mistreatment has not been carried out in the past.

This study extends research on mistreatment by patients to the domain of nurses’ job satisfaction and turnover intention, which are increasingly recognized as critical factors for employee retention [75]. The mistreatment literature predominantly focuses on employees’ emotional and retaliatory variables as the outcomes of customer mistreatment [6,76]. A key contention in mistreatment studies is that customer mistreatment enhances employees’ negative emotions [77]. We propose that, in the nursing context, this contention hinges on the critical role played by employee experiences of mistreatment by patients in reducing employees’ perceived work meaningfulness, the source of job satisfaction, as well as increasing employees’ emotional dissonance, the source of turnover intention, which was consistent with previous research[78], mistreatment by service receiver has negative influence on service provider’s emotions and actions.

Our study answers the calls of previous research to study and attest the importance of hostile attribution in the relationship between mistreatment by patients and work meaningfulness/emotional dissonance. Recent mistreatment research shows that the moderating effect of hostile attribution bias on the associations between mistreatment and emotional reactions is not yet documented [6]. In line with attribution theory[79], this study advances an attribution model on mistreatment by patients and demonstrates that it will have a weaker stronger negative effect on work meaningfulness (a stronger positive effect on emotional dissonance) when hostile attribution bias is high rather than low.

The current findings have valuable implications for nurse managers. Our study shows that mistreatment by patients is significant in lowering nurses’ job satisfaction and increasing their turnover intention while previous study showed that the safe work environments which are devoid of violence may enhance nurses’ job satisfaction [80]. One strategy would be to take steps to prevent mistreatment by patients from occurring in the first place. Hospitals might use signs to remind patients to treat nurses with dignity and respect and to institute a zero-tolerance policy toward mistreatment by patients [81]. This strategy might not only help to reduce mistreatment by patients but also show that hospitals care about treating nurses with dignity and respect.

Our findings indicate that hostile attribution bias moderates the relationship between mistreatment by patients and work meaningfulness/emotional dissonance, which also provides a new insight into how managers can approach this problem. Previous research has suggested that hostile attribution bias is changeable [50]. Managers should discourage nurses’ hostile attribution bias; it is plausible that when nurses attribute mistreatment to reasons other than hostility, there is no need to respond negatively. Moreover, Yagil and Dayan’s [82] findings extended extant research on the causes of mistreatment against nurses by highlighting a tendency to view certain circumstances as justifying such behavior, that if the nurse was not caring enough, or did not communicate clearly enough, the patient had the right to get aggressive. Or that the patient is in just great distress, enough to justify mistreatment which one needs to be trained to tolerate. There has to be a shift in how we perceive or tolerate or justify mistreatment, so that organizations and the public can be educated to embrace fundamental changes on what is acceptable behavior towards healthcare providers.

In addition, organizations can help nurses to deal with mistreatment by patients and associated resource loss. Previous research has suggested that an individual’s perceived social support at work can buffer the negative effects of mistreatment[82]. Further support, such as supervisor or organizational support, should be provided to nurses to help them gain and maintain resources to avoid emotional dissonance and the loss of work meaningfulness [83].

REFERENCES

Many bibliographies are obsolete. The bibliographic citations used are more than 5 years old (77,9%). The authors must update and arrange the bibliography.

Response:

Thank you for your suggestion. In the revised manuscript, we updated our references.

Reviewer 2 Report

This is an interesting paper on the job satisfaction elements in chinese nurses.

Research in this area is important and it would be helpful to the reader to have some clearer insight on what the authors mean by "mistreatment by patients" in terms of specific elements. As far as I can see there is only some brief reference in the Methods section lines 279-282. 

Have the authors considered the effect of personal perceptions, attitudes, important life-events of the participating nurses or social determinants like salary, employment prospects, or relationships with superiors at work on the characterization of events as "mistreatment" or "rough treatment"?

How have the authors managed to separate the effect of work events from events in perosnal life of the participants?

Was there any pilot phase or preparatory discussion with the study subjects to dissociate their answers from other non-work-related events?

This interconnection should be noted in the limitations section.

Finally, some language spelling and grammar should be performed, e.g. line 304-306 ...are likely to be associated, to control for age, gender...

Author Response

Response to Reviewer 2

Reviewer #2:

Comments and Suggestions for Authors

This is an interesting paper on the job satisfaction elements in chinese nurses.

Response:

Thank you for your encouragement.

Research in this area is important and it would be helpful to the reader to have some clearer insight on what the authors mean by "mistreatment by patients" in terms of specific elements. As far as I can see there is only some brief reference in the Methods section lines 279-282.

Response:

Thank you for your suggestion. We revised our manuscript and add all the items of mistreatment by patients in the method section. Specific as follows:

“Mistreatment by patients (T1). We adopted the seven-item mistreatment by patients scale from Cortina et al. [63]. Items include: “Patients doubted my judgment on a matter over which I have responsibility,” “Patients made unwanted attempts to draw me into a discussion of personal matters,” “Patients addressed me in an unprofessional term, either publicly or privately,” “Patients put me down or was condescending to me,” “ Patients paid little attention to my statement or showed little interest in my opinion,” “Patients made demeaning or derogatory remarks about me”, and “Patients yelled at me”. “Doubted my judgment on a matter over which I have responsibility,” “Made unwanted attempts to draw me into a discussion of personal matters,” and “Addressed me in an unprofessional term, either publicly or privately.” Cronbach’s alpha for this measure was 0.982.”

Have the authors considered the effect of personal perceptions, attitudes, important life-events of the participating nurses or social determinants like salary, employment prospects, or relationships with superiors at work on the characterization of events as "mistreatment" or "rough treatment"?

Response:

Thank you for your comment. The essence of mistreatment is a type of low-quality interpersonal treatment that nurses receive from their service receivers during service interactions (Wang et al., 2013), and often creates an unsafe work environment for nurses (Vagharseyyedin, 2016). Previous studies have already indicated that personal characteristics, job characteristics (i.e., job complexity, skill variety, job autonomy, and task significance), informal work environment (i.e., organizational support, service climate), organizational polices (i.e., display rules, service delivery), and individual states (i.e., state mood, state regulatory resources) would influence mistreatment (Koopmann et al., 2015).

In the current study, we aim to investigate what will nurses react when they suffered mistreatment, not what will influence mistreatment. Consider that employees’ attributions should be given attention in research on mistreatment by service users (Wang et al., 2013). It is plausible that nurses with different level of hostile attribution bias regarding mistreatment by patients, may react differently. Thus, wo introduce hostile attribution bias as a potential moderator to investigate the relationship between mistreatment by patients and nurses’ job satisfaction and turnover intention.

Reference:

Koopmann, J. , Wang, M. , Liu, Y. , & Song, Y. . (2015). Customer mistreatment: a review of conceptualizations and a multilevel theoretical model. Research in Occupational Stress and Well Being, 13, 33-79.

Wang, M., Liu, S., Liao, H., Gong, Y., Kammeyer-Mueller, J., Shi, J. (2013). Can't get it out of my mind: Employee rumination after customer mistreatment and negative mood in the next morning. Journal of Applied Psychology, 98(6), 989-1004.

How have the authors managed to separate the effect of work events from events in personal life of the participants?

Was there any pilot phase or preparatory discussion with the study subjects to dissociate their answers from other non-work-related events?

This interconnection should be noted in the limitations section.

Response:

Thanks very much for your question. We have two ways to separate the effect of work events from events in personal life of the participants:

First, the method section mentioned that, before delivering the questionnaires, we explained the purpose of the study to all the participants. That is to say, all the participants know that they are going to answer questions about their workplace.

Second, all the items of mistreatment by patients focused on “patients”, specific as follows:

Patients doubted my judgment on a matter over which I have responsibility” “Patients made unwanted attempts to draw me into a discussion of personal matters”

“Patients addressed me in an unprofessional term, either publicly or privately” “Patients put me down or was condescending to me”

“ Patients paid little attention to my statement or showed little interest in my opinion”

“Patients made demeaning or derogatory remarks about me”

“Patients yelled at me”  

Above two way may help separate the effect of work events from events in personal life of the participants.

Finally, some language spelling and grammar should be performed, e.g. line 304-306 ...are likely to be associated, to control for age, gender...

Response:

Thank you for pointing this out. In the revised manuscript, we corrected the spelling and grammar mistake and checked all the details carefully.

Reviewer 3 Report

Overall impression:

This is an important subject which has significant impact on the wellbeing of health care providers, which in turn affects clinical care of patients. It is well-written, with minimal need for changes, and has included relevant bibliography. There were no ethical concerns raised. The study makes many interesting points, but the conclusion still seems to focus on the need for the nurse to tolerate or change herself/himself by reducing hostile attribution bias. While hostile attribution bias can moderate the relationship between mistreatment by patients and work meaningfulness/emotional dissonance, it would be more helpful to put a greater emphasis on the need for organizations and hospitals to clearly advocate for zero tolerance of mistreatment of healthcare workers.

The following are some of my suggestions, which I hope will be helpful for the authors.

Introduction:

Line 24-25 refers to the Chinese Medical Occupation Status White Paper issued by the Chinese Medical Doctor Association in 2018 – I was unable to find the data quoted through internet searches. It is easy to find the Chinese Physicians’ Practice Status White Paper, but not the one referred to. It would be helpful to add a link to the cited paper.

Line 33 – “few have paid attention to…” likely refers to mistreatment by patients.

Discussion:

The discussion could include another aspect in aggression – how it may be justified by society, that if the nurse was not caring enough, or did not communicate clearly enough, the patient had the right to get aggressive. Or that the patient is in just great distress, enough to justify aggression which one needs to be trained to tolerate. There has to be a shift in how we perceive or tolerate or justify aggression, so that organizations and the public can be educated to embrace fundamental changes on what is acceptable behavior towards healthcare providers.

Yagil D, Dayan H: Justification of aggression against nurses: The effect of aggressor distress and nurse communication quality. J Adv Nurs. 2020 Feb;76(2):611-620.

From this perspective, one could argue that it does not make sense to advocate for line 423-425: “hospitals can provide mentoring and training programs that present inconsistent information to facilitate nurses’ understanding that not all mistreatment by patients is intentional, especially when heavy workloads lead nurses to neglect others”.

Author Response

Response to Reviewer 3

Dear Editor and Reviewers,

Thanks very much for providing us with an opportunity to revise and resubmit our manuscript. We sincerely thank you for your kind and helpful comments. We have deeply considered each of the comments and carefully revised the manuscript. In this revised version, we hope our major changes can significantly improve our manuscript and thus meet your expectations. For your convenience, we provide specific response to each of the comments by the reviewers below.

Reviewer #3:

Comments and Suggestions for Authors

Overall impression:

This is an important subject which has significant impact on the wellbeing of health care providers, which in turn affects clinical care of patients. It is well-written, with minimal need for changes, and has included relevant bibliography. There were no ethical concerns raised. The study makes many interesting points, but the conclusion still seems to focus on the need for the nurse to tolerate or change herself/himself by reducing hostile attribution bias. While hostile attribution bias can moderate the relationship between mistreatment by patients and work meaningfulness/emotional dissonance, it would be more helpful to put a greater emphasis on the need for organizations and hospitals to clearly advocate for zero tolerance of mistreatment of healthcare workers.

Response:

Thank you for your encouragement and suggestions. We have carefully thought every piece of your advice and try our best to solve these issues, hoping to improve our manuscript and meet your requirements. In the revised manuscript, we added some practical implications in the discussion section, specific as follows:

“Our findings indicate that hostile attribution bias moderates the relationship between mistreatment by patients and work meaningfulness/emotional dissonance, which also provides a new insight into how managers can approach this problem. Previous research has suggested that hostile attribution bias is changeable [50]. Managers should discourage nurses’ hostile attribution bias; it is plausible that when nurses attribute mistreatment to reasons other than hostility, there is no need to respond negatively. Moreover, Yagil and Dayan’s [82] findings extended extant research on the causes of mistreatment against nurses by highlighting a tendency to view certain circumstances as justifying such behavior, that if the nurse was not caring enough, or did not communicate clearly enough, the patient had the right to get aggressive. Or that the patient is in just great distress, enough to justify mistreatment which one needs to be trained to tolerate. There has to be a shift in how we perceive or tolerate or justify mistreatment, so that organizations and the public can be educated to embrace fundamental changes on what is acceptable behavior towards healthcare providers.”

The following are some of my suggestions, which I hope will be helpful for the authors.

Introduction:

Line 24-25 refers to the Chinese Medical Occupation Status White Paper issued by the Chinese Medical Doctor Association in 2018 – I was unable to find the data quoted through internet searches. It is easy to find the Chinese Physicians’ Practice Status White Paper, but not the one referred to. It would be helpful to add a link to the cited paper.

Response:

Thank you for your suggestion. The data were form the Chinese Medical Occupation Status White Paper, not any paper. In the revised manuscript, we added the URL as foot note:

http://www.xinhuanet.com/gongyi/2017-05/11/c_129601688_2.htm

Line 33 – “few have paid attention to…” likely refers to mistreatment by patients.

Response:

Thank you for pointing out that. We revised this sentence to make it more clear, specific as follows:

Though many studies have explored mistreatment in the workplace, most concern customer mistreatment; few have paid attention to mistreatment by patients.

Discussion:

The discussion could include another aspect in aggression – how it may be justified by society, that if the nurse was not caring enough, or did not communicate clearly enough, the patient had the right to get aggressive. Or that the patient is in just great distress, enough to justify aggression which one needs to be trained to tolerate. There has to be a shift in how we perceive or tolerate or justify aggression, so that organizations and the public can be educated to embrace fundamental changes on what is acceptable behavior towards healthcare providers.

Yagil D, Dayan H: Justification of aggression against nurses: The effect of aggressor distress and nurse communication quality. J Adv Nurs. 2020 Feb;76(2):611-620.

From this perspective, one could argue that it does not make sense to advocate for line 423-425: “hospitals can provide mentoring and training programs that present inconsistent information to facilitate nurses’ understanding that not all mistreatment by patients is intentional, especially when heavy workloads lead nurses to neglect others”.

Response:

Thank you for your suggestion. In the revised manuscript, we added a new aspect according to your suggestion, specific as follows:

“Our findings indicate that hostile attribution bias moderates the relationship between mistreatment by patients and work meaningfulness/emotional dissonance, which also provides a new insight into how managers can approach this problem. Previous research has suggested that hostile attribution bias is changeable [50]. Managers should discourage nurses’ hostile attribution bias; it is plausible that when nurses attribute mistreatment to reasons other than hostility, there is no need to respond negatively. Moreover, Yagil and Dayan’s [79] findings extended extant research on the causes of mistreatment against nurses by highlighting a tendency to view certain circumstances as justifying such behavior, that if the nurse was not caring enough, or did not communicate clearly enough, the patient had the right to get aggressive. Or that the patient is in just great distress, enough to justify mistreatment which one needs to be trained to tolerate. There has to be a shift in how we perceive or tolerate or justify mistreatment, so that organizations and the public can be educated to embrace fundamental changes on what is acceptable behavior towards healthcare providers.”

Reviewer 4 Report

Before publication the work needs improvement, especially the methodology section.
No relevant information regarding the conduct of the study.

Thank you very much for the opportunity to review this article.
The work deals with the very important topic of bad treatment of medical personnel, especially nurses by patients, however, I ask the Authors to explain some issues, especially those regarding the methodology of the research.

Comments to the Authors:
- Abstract - no clearly defined aim of the study.
Sentence: “We expected the proximal……….”, from line 13 to 16 is very long and difficult to follow.
Keywords are more phrases than words. It needs to be changed and sorted in alphabetical order.
Methodology section:
The authors write that the study was carried out in three hospitals, why in three, not four, why exactly in these.
On what basis the hospitals were selected?
Lack of information about the consent of the facility's director.
On line No 248, what does initial contact mean?
The authors write about a well-randomized study group, what was this random selection?
How were the nurses recruited for the study?
Please enter some inclusion and exclusion criteria.
What were the nurses, what departments?
How much time did it take for researchers to collect data, how did we manage to meet such a large group during typical working- day – please accurately describe the organization of the study.
In line 255, the authors write: “To alleviate the potential for common method bias concern”,
what do the authors mean by this?
The authors write about coding surveys, how the surveys were coded, since the survey was conducted three times, how it was taken care to link the results of individual studies with maintaining the anonymity of the respondents.
Please describe what statistical methods were used and whether these methods were followed.
There is no information about obtaining the consent of the Bioethics Committee. Have the authors obtained such consent, if so please provide the number.
No information on the availability of data, please attach a link to the repository where such data can be found.
Line 267-270: there are the results and this information should be included in the results section as the characteristics of the study group.
By the way, is very little information about the nurses surveyed, only age and average seniority?
What about education, type of work, additional qualifications?
These are factors that can call for better life satisfaction
Questionnaires used: did the authors carry out only language or cultural adaptation?
Have these questionnaires been validated for your country?
Have the authors obtained permission to use them?
Maybe you can include questionnaire templates as additional files for review?
The authors state that they paid research to nurses for completed surveys, did this not affect the bias of the study?
Before publication, the work requires a thorough correction.

Author Response

Response to Reviewer 4

Dear Editor and Reviewers,

Thanks very much for providing us with an opportunity to revise and resubmit our manuscript. We sincerely thank you for your kind and helpful comments. We have deeply considered each of the comments and carefully revised the manuscript. In this revised version, we hope our major changes can significantly improve our manuscript and thus meet your expectations. For your convenience, we provide specific response to each of the comments by the reviewers below.

Reviewer #4:

Comments and Suggestions for Authors

Before publication the work needs improvement, especially the methodology section.

No relevant information regarding the conduct of the study.

Thank you very much for the opportunity to review this article.

The work deals with the very important topic of bad treatment of medical personnel, especially nurses by patients, however, I ask the Authors to explain some issues, especially those regarding the methodology of the research.

Response:

Thank you for your suggestions. We have carefully thought every piece of your advice and try our best to solve these issues, hoping to improve our manuscript and meet your requirements.

Comments to the Authors:

- Abstract - no clearly defined aim of the study.

Sentence: “We expected the proximal……….”, from line 13 to 16 is very long and difficult to follow.

Keywords are more phrases than words. It needs to be changed and sorted in alphabetical order.

Response:

Thank you very much for your comments. We revised the abstract section, specific as follows:

“The affective event of mistreatment in the workplace has been recognized as an important factor influencing employee affect and behavior. However, few studies have logically explained and empirically clarified the link between mistreatment by patients and nurses’ job satisfaction and turnover intention. The current study aimed to explore the effects of mistreatment by patients on nurses’ job satisfaction and turnover intention through work meaningfulness and emotional dissonance, as well as the moderating role of hostile attribution bias. Using three-wave survey data collect from 657 nurses who worked in three hospitals in China, we found that mistreatment by patients had a negative effect on nurses’ job satisfaction through work meaningfulness, mistreatment by patients had a positive effect on nurses’ turnover intention through emotional dissonance. Furthermore, nurses’ hostile attribution bias acted as an effective moderator on such relationship These findings help uncover the mechanisms and conditions in which mistreatment by patients influences nurses’ job satisfaction and turnover intention.”

Furthermore, all the key words are phrase and ranked by importance.

The authors write that the study was carried out in three hospitals, why in three, not four, why exactly in these.

On what basis the hospitals were selected?

Response:

Thank you for this comment. The reason why we conducted the study in these three hospitals as follows:

First, hospitals in China are organized according to a 3-tier system that recognizes a hospital's ability to provide medical care, medical education, and conduct medical research. Tertiary hospitals round up the list as comprehensive or general hospitals at the city, provincial or national level with a bed capacity exceeding 500. All these three hospitals are tertiary hospitals, there are a large number of nurses that can be investigated.

Second, only these hospitals’ top managers allow us to do the survey in their hospitals.

Third, we also considered the principle of data availability.

Thus, we chose these three hospitals, not four or more.

Lack of information about the consent of the facility's director.

Response:

Thank you for this comment. In the revised manuscript, we added this information, specific as follows:

“Nurses from three hospitals (two in Jinan and one in Taiyuan, China) participated in the survey. Hospital managers and human resource supervisor authorized this survey. We confirmed that each respondent is willing to accept the questionnaire survey before we delivery the questionnaires.”

On line No 248, what does initial contact mean?

Response:

Thank you for your question. We mean before conducting the survey. In the revised manuscript, we delete this sentence to avoid misunderstanding.

The authors write about a well-randomized study group, what was this random selection?

How were the nurses recruited for the study?

Please enter some inclusion and exclusion criteria.

What were the nurses, what departments?

Response:

Thank you for your questions.

For the first question: Random sampling method is also called "sampling survey method". In statistics, a simple random sample is a subset of individuals (a sample) chosen from a larger set (a population). Each individual is chosen randomly and entirely by chance, such that each individual has the same probability of being chosen at any stage during the sampling process, and each subset of k individuals has the same probability of being chosen for the sample as any other subset of k individuals (Yates et al., 2008). Before distributing questionnaires, we contacted with these hospitals’ managers and got names of nurses from HR. Then, we randomly selected 1200 nurses from theses names and coded each nurse as T11-T11200 (T1 stands for time 1), T21-T21200 (21 stands for time 2), T31-T31200 (T3 stands for time 3). When distributing the questionnaire, we used numbers we coded instead of names.

For the second question: We first contacted the top managers of hospitals; and there are three hospitals located in Jinan and Taiyuan agreed to cooperate with our study. Then, we contacted the human resource departments of these hospitals. The study announcement, along with a letter assuring confidentiality and voluntary participation, was distributed to them by the human resource department. Finally, 1200 nurses who were randomly selected expressed an understanding of the intention and purpose of the study and consented to participate.

For the third question: there’s no inclusion and exclusion criteria, we don’t know any nurses before conducting the survey, just randomly selected participants from a large group.

For the fourth question: the nurses are from all departments that contact patients directly.

Reference:

Yates, Daniel S.; David S. Moore; Daren S. Starnes (2008). The Practice of Statistics, 3rd Ed. Freeman. ISBN 978-0-7167-7309-2.

How much time did it take for researchers to collect data, how did we manage to meet such a large group during typical working- day – please accurately describe the organization of the study.

Response:

Thank you for your questions. It took us almost three months to collect data (from November 2016 to January 2017). We recruited ten research assistant to help us collect data. Before conducting this survey, we explained the purpose of the study, assured confidentiality, and emphasized the importance of honest answers for scientific research before distributing the survey questionnaires. We confirmed that each respondent is willing to accept the questionnaire survey. We got every participant’s work time from HR and asked he/she to finish these questionnaires in the hospital after he/she getting off work before going home. We took back the completed questionnaire directly on the spot.

In line 255, the authors write: “To alleviate the potential for common method bias concern”, what do the authors mean by this?

Response:

Thank you for your comment. It is widely assumed that common method bias inflates relationships between variables measured by self-reports (Conway & Lance, 2010). Podsakoff and Todor (1985) stated ‘‘Invariably, when self-report measures obtained from the same sample are utilized in research, concern over same-source bias or general method variance arises’’ (p. 65). Subsequently, Podsakoff et al. (2003) proposed techniques for controlling common method bias, one is to create a temporal separation by introducing a time lag between the measurement of the predictor and criterion variables. According above, we conducted a time-lagged three-wave design to reduce common method bias for the data were collected from the same source.

Reference:

Conway, J. M. & Lance. C. E.. (2010). What reviewers should expect from authors regarding common method bias in organizational research. , 25(3), 325-334.

Podsakoff, P. M., Mackenzie, S. B., Lee, J. Y., & Podsakoff, N. P. (2003). Common method biases in behavioral research: a critical review of the literature and recommended remedies. Journal of Applied Psychology, 88(5), 879-903.

Podsakoff, P. M., & Todor, W. D.. (1985). Relationships between leader reward and punishment behavior and group processes and productivity. Journal of Management, 11, 55–73.

The authors write about coding surveys, how the surveys were coded, since the survey was conducted three times, how it was taken care to link the results of individual studies with maintaining the anonymity of the respondents.

Response:

Thank you for your questions. Before distributing questionnaires, we contacted with these hospitals’ managers and got names of nurses from HR. Then, we randomly selected 1200 nurses from theses names and coded each nurse as T11-T11200 (T1 stands for time 1), T21-T21200 (21 stands for time 2), T31-T31200 (T3 stands for time 3). When distributing the questionnaire, we used numbers we coded instead of names. We took back the completed questionnaire directly on the spot, assured confidentiality.

Please describe what statistical methods were used and whether these methods were followed.

Response:

Thank you for your comment. In the manuscript, we added data analysis in the method section, specific as follows:

“3.3. Data Analysis

We used SPSS 23.0 to implement reliability and descriptive statistics and employed Mplus 7.4 to implement CFA before testing our hypotheses. We used a bootstrapping approach, including 95% confidence intervals (CI) using 2000 bootstrap samples in Mplus 7.4 to test our hypotheses.”

There is no information about obtaining the consent of the Bioethics Committee. Have the authors obtained such consent, if so please provide the number.

Response:

Thank you for your comment. An ethics approval was not required as per institutional guidelines and national laws regulations because there’s no unethical behaviors existing in the research procedures. We just conducted questionnaire survey and were exempt from further ethics board approval since our research did not involve human clinical trials or animal experiments. Also, the content of the questionnaire did not involve any sensitive or personal privacy or ethical and moral topics. In the first page of the questionnaire, information on consent procedures was included and participants were notified that consent was to be obtained by virtue of survey completion. Meanwhile, we informed that participants about the objectives of the study and guaranteed their confidentiality and anonymity. All the participants were completely free to join or drop out the survey. Only those who were willing to participate were recruited.

No information on the availability of data, please attach a link to the repository where such data can be found.

Response:

Thank you for your comment. According to the journal submission requirements, we did not provide the data. In the revised manuscript, we added one sentence in the method section, If anyone want to get the research data can contact us.

Line 267-270: there are the results and this information should be included in the results section as the characteristics of the study group.

Response:

Thank you for your comment. According to the structure of the journal articles, we put the characteristics of the study group in the method section.

By the way, is very little information about the nurses surveyed, only age and average seniority? What about education, type of work, additional qualifications?

These are factors that can call for better life satisfaction

Response:

Thank you for your comment. According to previous research, employee demographics are likely associated correlated with nurses’ job satisfaction (Li & Lambert, 2008) and turnover intention (EmiroÄźlu, Akova, Tanrıverdi, 2015), we thus controlled for gender, age, and organizational tenure for their potential effects. Of course we agree with your opinion there are many other factors like education, type of work, and additional qualifications may influence nurses’ job satisfaction and turnover intention. In the revised manuscript, we added this issue in the section of limitations and future research directions. Specific as follows:

Although we took nurses’ gender, age, organizational tenure as controls, there are many other factors like education, type of work, and additional qualifications may influence nurses’ job satisfaction and turnover intention. We call for future research control more factors for their potential effects.”

Questionnaires used: did the authors carry out only language or cultural adaptation?

Have these questionnaires been validated for your country?

Have the authors obtained permission to use them?

Maybe you can include questionnaire templates as additional files for review?

The authors state that they paid research to nurses for completed surveys, did this not affect the bias of the study?

Response:

Thank you for your questions.

For the first question: We followed the back-translation procedure recommended by Brislin to translate the measures (Brislin, 1970). A management professor fluent in both English and Chinese translated the items from English to Chinese, and then another bilingual management professor translated them from Chinese back into English. Discrepancies in the translation were resolved through discussion. A few participants from the participating hospitals were consulted to guarantee that the items could be generalized to the study context (Schaffer & Riordan, 2003).

For the second question: all the items of this research constructs have been used in China.

For the third question: questionnaires were made by research constructs’ items, all these constructs’ items are permitted to use.

For the fourth question: see Appendix for questionnaire templates.

For the fifth question: We paid money to nurses for completing surveys wouldn’t affect the bias of the study. First, we explained the purpose of the study, assured confidentiality, emphasized the importance of honest answers for scientific research before distributing the survey questionnaires. Second, most of the empirical research will pay money to participants for completing questionnaires, it is a useful way to improve questionnaire response rate.

Before publication, the work requires a thorough correction.

Response:

Thank you for your comments and suggestions. We have carefully thought every piece of your advice and try our best to solve these issues, hoping to improve our manuscript and meet your requirements.

Round 2

Reviewer 4 Report

Thank the Authors for improving the article.

The authors replied: "In the revised manuscript, we added one sentence in the method section, If anyone want to get the research data can contact us"
Unfortunately, I did not find this information, maybe I missed, if not, please add .

To my knowledge, despite the consent of the Bioethical Committee is required even in the case of surveys.
I leave this issue to the Editor's decision.